# Recent changes in extreme wave events in the Southwestern South Atlantic

Carolina B. Gramcianinov[1], Joanna Staneva[1], Celia R. G. Souza[2,3], Priscila Linhares[3], Ricardo de Camargo[4], and Pedro L. da Silva Dias[4]

[1]Institute for Coastal Systems Analysis and Modeling, Helmholtz-Zentrum Hereon, Max-Planck-Strasse 1, 21502 Geesthacht, Germany
[2]Institute of Environmental Research, Secretariat of Environment, Infrastructure and Logistics of São Paulo State (SEMIL/SP), Rua Joaquim Távora 822, 04015-011, São Paulo, SP, Brazil.
[3]Department of Physical Geography, Faculty of Philosophy, Literature and Human Sciences, University of São Paulo (FFLCH/USP), Av. Prof. Lineu Prestes, 338, 05508-000, São Paulo-SP, Brazil
[4]Department of Atmospheric Sciences, Institute of Astronomy, Geophysics and Atmospheric Science, University of São Paulo, Rua do Matão, 1226, 05508-090, São Paulo-SP, Brazil

**Correspondence:** C. B. Gramcianinov (carolina.gramcianinov@hereon.de)

**Abstract.** Over the past decades, the South Atlantic Ocean has experienced several changes, including a reported increase in coastal erosion and floods. This study aims to investigate the recent changes in the extreme wave events over the Southwest South Atlantic (SWSA) – which hosts the most economically important harbours in South America, high oil and gas production demands, and rich biodiversity. This investigation considers not only the occurrence of extreme wave events but also extreme
wave indicators that characterise the potential wave impact on offshore and coastal areas. Extreme wave events are obtained using the averaged monthly $95^{th}$ percentile of significant wave height ($Hs$) from 1993 to 2021, combining the CMEMS global wave reanalysis and near-real-time products. Annual and seasonal statistics are performed to analyse mean and extreme wave climate and trends in the study region, focusing on $Hs$, peak period, and wave power. The analysis gives an overview of the wave climate in the study domain, including the discussion about seasonal differences. For a more direct application to future
risk assessment and management, we perform an analysis considering the regional monitoring and warning system division established by the Brazilian Navy. We used a coastal hazards database that covers a portion of the coast to investigate how the trends given by the CMEMS wave products may impact the coastal zone. Our findings showed significant changes in the SWSA mainly associated with an increase in mean values of $Hs$, wave period, and, consequently, the wave power. Narrowing down to the coast impact, we found an increase in the number of coastal hazards in São Paulo State associated with waves,
which agrees with the increase in the number of extreme wave events in the adjacent ocean sector. However, the increased coastal events are also driven by local factors.

**Short Summary**

We analysed the extreme wave events trends in the Southwest South Atlantic in the last 29 years using wave products and coastal hazards records. The results showed important regional changes associated with increased mean sea wave height, wave

period and wave power. We also found a rise in the number of coastal hazards related to waves affecting the São Paulo State, which partially agrees with the increase in extreme waves in the adjacent ocean sector but is also driven by local factors.

## 1 Introduction

In recent years, several extreme events have been reported in the South Atlantic Ocean (e.g., Marcello et al., 2018; Dalagnol et al., 2022), thus reflecting directly on hazards along the coast. One of the regions that are facing relevant changes is the Southwestern South Atlantic (SWSA), with an increase in extreme wave and storm surges occurrence (Souza et al., 2019; Gramcianinov et al., 2022). With high economic relevance, the SWSA region hosts strategic harbours in South America, where 755 million tons of goods were transported in 2021 (ANTAQ, 2022) and promising oil and gas exploration fields. In addition, the region also holds rich biodiversity, including coral reefs and 856 $km^2$ of mangroves that are crucial for coastal hazard protection, economic activities (e.g., fishery) and the cultural identity of the coastal communities (ICMBio, 2018; Pereira-Filho et al., 2021). The SWSA coastal cities have a dense population, with approximately 20 million people who are extremely vulnerable to coastal erosion and infrastructure damage (Zamboni and Nicolodi, 2008).

Assessing the extreme waves and wave trends in the SWSA with traditional approaches has proven to be challenging for several reasons. The difficulties remain mostly in the still-limited understanding of the local physical processes (e.g., wave-current interaction) and climate variabilities (e.g., the overlapping effect teleconnections). The limited accuracy of long-term integrations and the scarce data availability can grieve these analyses even more. Some recent studies have revealed changes in the wave pattern in the South Atlantic, usually addressed to the increase in the extreme waves in the Southern Ocean (SO). In general, previous global or hemispheric-based studies have reported increases in wave height extremes in the Southern Hemisphere (SH) over the past 41 years, and this tendency is expected to continue in the future (Caires and Sterl, 2005; Dobrynin et al., 2012; Lemos et al., 2019). However, when focusing on the SWSA, the mean and extreme wave climate trends present larger uncertainties.

In addition to understanding the significant wave height ($Hs$) trends, assessing changes in wave event characteristics, such as the mean wave direction and peak period, is of utmost importance. Silva et al. (2020) showed how the oscillation between the south and east dominant wave energy flux directions has led to changes in the coastal morphodynamics at both regional and local scales. Some previous works reported wave power changes under the present climate (Odériz et al., 2021, 1979–2018) and mean wave direction and period changes in both present (Hemer et al., 2010, 1989-2005) and future climate (Lobeto et al., 2021, 2081-2100). These changes directly affect naval and coastal risk assessments, requiring special efforts to properly link the global scale findings to regional and local wave extremes.

Under such background, this section aims to report and investigate the recent extreme wave climate trends (1993 - 2021) in the SWSA while focusing on wave event characteristics such as events frequency, intensity, duration, and peak period. We examined the seasonal statistics and climatic trends using both traditional (i.e., percentile-based) and storm-based approaches to provide new insights into the regional wave climate changes. To obtain results with more direct application to future risk

assessment and management, we performed an analysis considering the regional monitoring and warning system, as well as the impact of the recent wave climate changes on the coast.

## 2 Methods

### 2.1 Datasets

The main dataset used in this work was the Copernicus Marine Service (CMEMS) global reanalysis, named WAVERYS (product ref. no. 1, Table 1; Law-Chune et al., 2021), available from 1993 to 2020. To include 2021 in the analysis, the WAVERYS was complemented with data from the CMEMS Global Ocean Waves Analysis Near Real-Time product (GLO-NRT; product ref. no. 2, Table 1). The combination (in time) of these two products is referred hereafter as CMEMS wave products. WAVERYS is available at a 1/5° horizontal grid as 3-hourly outputs from 1993 to 2020 while the wave analysis has a 1/12° horizontal grid as 3-hourly instantaneous output fields. The GLO-NRT product was interpolated to the 0.20° horizontal grid, so a more consistent analysis can be achieved despite using different sources. Both products are produced using Météo France Wave Model (MFWAM) with the dissipation terms developed by Ardhuin et al. (2010). WAVERYS is forced by hourly surface winds and daily the sea-ice fraction fields derived from the $5^{th}$ generation reanalysis from the European Centre for Medium-Range Weather Forecast (ECMWF) (ERA5; Hersbach et al., 2020) and ocean currents obtained from the ocean reanalysis Global Ocean Reanalysis and Simulation (GLORYS). The GLO-NRT is forced only by a 6-hourly winds analysis from the IFS-ECMWF atmospheric system.

An evaluation of WAVERYS for the western South Atlantic wave climate was made by Crespo et al. (2022). The authors compared $Hs$ from the WAVERYS, ERA5, and the National Center for Environmental Prediction (NCEP) Wave reanalysis (Chawla et al., 2013) against wave buoy measurements at three locations along the Brazilian coast and found that WAVERYS presented the highest correlation and the lowest root mean square deviation (RMSD). The ERA5 performance in representing the winds is also relevant once the quality of the forcing field is crucial in a wave simulation. Previous works have shown that ERA5 can represent the wind climate, extreme percentiles, and storm variability (e.g., Belmonte Rivas and Stoffelen, 2019; Gramcianinov et al., 2020a; Crespo et al., 2022).

**Table 1.** CMEMS and non-CMEMS products used in this study, including the Quality Information Document (QUID) and Product User Manual (PUM).

| Product ref. no. | Product ID & type | Data access | Documentation |
|---|---|---|---|
| 1 | GLOBAL_MULTIYEAR_WAV_001_032 (WAVERYS); Numerical models | (EU Copernicus Marine Service Product, 2021) | QUID: Law-Chune et al. (2021) PUM: Law-Chune (2022) |
| 2 | GLOBAL_ANALYSIS_FORECAST_WAV_001_027 (GLO-NRT); Numerical models | (EU Copernicus Marine Service Product, 2022) | QUID: Aouf (2022) PUM: Dalphinet et al. (2022) |
| 3 | Baixada Santista Coastal Hazards database (BDe-BS), data set | personal contact | Souza et al. (2019) Souza et al. (2022) |

## 2.2 Percentile computation

In this work, the percentiles were computed using the empirical distribution of the $Hs$ peaks ($Hs_{peaks}$) within a given period, thus allowing us to obtain a more detailed view of individual wave events' occurrence. The selected $Hs_{peaks}$ must be separated by a minimum of 48 hours to guarantee the independence of the peaks. This time window has been widely applied in past studies to ensure the collection of one peak per storm (e.g., Caires and Sterl, 2005; Meucci et al., 2020). Besides that, 48 hours is a suitable but not-so-restrictive time threshold for extreme wave analysis in the region, particularly considering the differences among the seasons. The $95^{th}$ percentile was computed based on the monthly $Hs_{peaks}$ distribution at each grid point. Using these monthly $95^{th}$ percentiles, we calculated the annual and seasonal means to analyze trends and proceed with the wave event analysis (Section 2.4). The seasonal mean of the $95^{th}$ monthly percentiles was computed for the summer and winter, using the average December-January-February (DJF) and June-July-August (JJA), respectively, thus having one value per year. The annual percentiles were computed by the average of all monthly percentiles within the year. As a final result, we have a mean annual and seasonal percentile time series at each grid point.

## 2.3 Trends estimation and testing

Trends were estimated based on Sen's slope estimator (Sen, 1968), which evaluates the magnitude of a time series trend. The significance of Sen's slope was calculated by the Mann-Kendall test (Mann, 1945; Kendall, 1975), considering a p-value lower than 0.05. Both methods are non-parametric (distribution-free) procedures and consider the monotonic upward or downward of the time series, thus, being more robust to climate-based analysis (e.g., Wang et al., 2020).

## 2.4 Extreme wave event analysis

The wave event statistics were derived following the methods developed by Weisse and Günther (2007), in which consecutive points over a specific threshold within a given time series are considered to define extreme wave events. This event-counting process was performed for each grid cell considering its unique severe event threshold (SET), defined herein as the average of the monthly $95^{th}$ percentile of $Hs_{peaks}$ considering the whole period (1993 - 2021). Notably, there is no widely accepted method for selecting threshold values and values between the $90^{th}$ and $99^{th}$-percentile $Hs$ are often used (Leo et al., 2020; Gramcianinov et al., 2020b). Moreover, the use of averaged monthly percentile results in a smoothed field, especially due to the $Hs_{peaks}$ variability among the year. In this way, for some locations, the exceedance of events above SET was large than 5%.

Following Weisse and Günther (2007), the intensity is equal to the difference between the maximum $Hs$ of the event and the SET at that point. The wave event statistics, such as the number of events, intensity, mean wave direction, and peak period, are presented herein as annual and seasonal means to build the spatial distribution and trends and obtain the spatial-averaged time series. The intensity and wave parameters were calculated by averaging all individual events (above SET) within the year or season. The same analysis was applied successfully in the Black Sea by Staneva et al. (2022), allowing a better understanding

of the extreme wave events' spatial distribution and trends. More details about these methods can be found in Weisse and Günther (2007).

## 2.5 Wave power calculation

Following Staneva et al. (2022), we also calculated the wave power in the study domain. Wave power or wave energy flux was obtained following the Eq. 1:

$$P = (\frac{\rho\,g^2}{64\,\pi})\,Hs^2\,Te,\tag{1}$$

where $P$ is the wave energy flux per unit of wave-crest length (kW/m), $\rho$ is water density, $g$ is the acceleration due to gravity, $Hs$ is the significant wave height, and $Te$ is the wave energy period. The $Te$ is directly available in the WAVERYS products (named VTM10) and is defined as the mean wave period obtained by the $Te \equiv Tm_{-1,0} = m_{-1}/m_0$, based on the $-1th$ and $0th$ moment of the wave spectrum.

## 2.6 Coastal risk assessment

Warnings and risk assessment in this region are supervised by the Center of Hydrography of the Brazilian Navy (CHM, from "Centro de Hidrografia da Marinha"), which is recognised by the World Meteorological Organization (WMO) as the issue service for the MetArea V (Atlantic waters west of 20°W, 7°N - 35.8°S). According to the CHM monitoring system, the coastal region of SWSA can be divided into 4 subareas (Fig. 1). These subareas were used to analyze the wave climate trends in the domain, considering the regional specificity and facilitating future discussions about risk management. This analysis may facilitate the applicability of the results found here to improve future monitoring and warning system development.

We used a historical database of coastal hazards in São Paulo State, within the C subarea, to further investigate coastal impacts. The Baixada Santista Coastal Hazards database (BDe-BS, product ref. no. 3, Table 1) covers the period from 1928 to 2021 and is obtained using the hemerographic method (mostly newspapers) and material from social media (mostly videos), showing coastal impacts caused by strong waves and anomalous high tides (either meteorological or astronomical tides) (Souza et al., 2019; Linhares et al., 2021; Souza et al., 2022). The definition of coastal hazards is mainly based on processes such as coastal erosion and/or coastal inundation, the latter also forced by continental flooding (heavy rainfall) in estuarine areas. Therefore, the coastal hazards registered in the BDe-BS represent events with high intensity since they were brought to the attention of the public due to their significant impact on the beaches, destruction of urban structures, and public and private properties, as well as disruption of the city's day-to-day and port activities. More details regarding the database can be found in Souza et al. (2019).

Currently, the BDe-BS initiative is maintained by the São Paulo State government through the "Preventive Plan for Coastal Erosion, Coastal Inundation and Flooding" (adapted from the Portuguese: "Plano Preventivo de Defesa Civil para Erosão Costeira, Inundações Costeiras e Enchentes/Alagamentos causadas por Eventos Meteorológicos-Oceanográficos Extremos como Ressacas do Mar e Marés Altas"). Despite representing a small portion of the coastal area of the SWSA, the number of

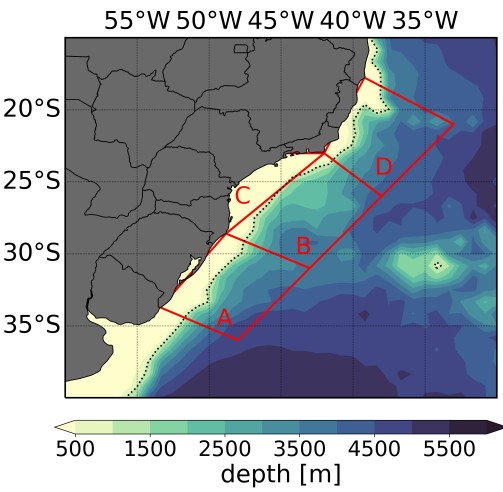

**Figure 1.** The Southwestern South Atlantic Ocean with the A to D subareas defined by the Brazilian Navy for warning and monitoring operations within MetArea V. Shaded values are the bathymetry from ETOPO1.

intense/extreme events reaching the São Paulo State coast can be considered representative of most of the coastal extension of the domain in this study, except for the subarea D (see Fig. 1). The reason for this extrapolation is mainly the lack of long-term records in other locations. In addition, the database covers the central portion of the study region in a region with high economic importance.

## 3 Results

### 3.1 Extreme wave climate characterization

The $95^{th}$ percentile distribution with its gradient towards the south (Fig. 2a,e,i) followed by the concentration of more intense extreme events to the southern portion of the domain (Fig. 2c,g,k) reflects the influence of the South Atlantic storm track in the region. The storm track controls the extreme wave climate in the SWSA due to the strongest winds associated with the cyclones (e.g., Campos et al., 2018) and it is located between 55°S and 30°S (Hoskins and Hodges, 2005; Gramcianinov et al., 2019).

Despite sharing a common extreme wave-generation source, as it is possible to see by the similar mean wave direction distribution (Fig. 2d,h,l), summer (DJF) and winter (JJA) present distinct wave patterns due to the southward shift of the storm track in the summertime (Hoskins and Hodges, 2005). Thus, the summer presents smaller values of $Hs$ - and consequently, $95^{th}$ percentile $Hs_{peaks}$ ($Hs_{p95}$ hereafter) values (Fig. 2e) - in the study domain, which reflects in a lower number of events (Fig. 2f) and weaker events (Fig. 2g) than winter and the whole period. Climatologically, the austral autumn (MAM) presents

behaviour closer to the summer pattern, while spring (SON) and winter patterns are similar. In this study, we will analyse further the whole period and winter since summer does not present many cases.

During the winter, the main storm track is in its northernmost position (Hoskins and Hodges, 2005), resulting in more wave events than in other seasons (Fig. 2j). Typically, in winter, the region presents relatively long fetches along the coast (southwest/northeast orientated) associated with cyclones generated at approximately 35°S (Gramcianinov et al., 2021). These fetches can be widely intensified by rear anticyclones on the western side of the cyclone, thus causing this configuration to be widely related to the most severe cases observed in the domain (e.g., da Rocha et al., 2004; Machado et al., 2010; Dragani 160 et al., 2013).

    The high number of events in the northern boundary of the domain (Fig. 2b,j) can be associated with the South Atlantic Subtropical High (SASH), which is also a generating system in the study region (Pianca et al., 2010). The SASH influences mostly the wave climate by generating easterly waves towards the central Brazilian, northward from 23°S. However, the wave events in this location are associated with relatively small $Hs$, as it is possible to see by the local $Hs_{p95}$ values (Fig. 2a,i) and 165 wave events intensities (Fig. 2c,k). For instance, the $Hs_{p95}$ values in the northern portion of the domain do not reach 3.5 m in the winter (Fig. 2i).

    The overall pattern and values presented in Fig. 2 agree with previous studies, even though methodological differences exist, thus making a straightforward comparison difficult. For instance, Gramcianinov et al. (2020b), using the $90^{th}$ percentile computed through a spatially-varying time window, found a mean of 1.3 and 5.5 extratropical cyclones per year associated 170 with extreme waves events in the region in the summer and winter, respectively. These values are comparable with the number of events presented in the maps of Fig. 2b,g,l. Regarding the intensity, the same authors found the mean $Hs$ of 6.5 m associated with these cyclones' events, which is also comparable to the intensity values (above the percentile) in some locations of the study domain (Fig. 2c,h,m). Moreover, Machado et al. (2010) accessed extreme wave events in the coastal region between 30°S and 32°S and found a mean of 1.33 events per year above the $90^{th}$ percentile between 1979 and 2008. We also reported 175 this relatively small value at this exact location in Fig. 2b,g,l. In this way, the method applied herein presents robust results according to what is reported in the region.

### 3.2   Extreme wave events trends

The monthly $Hs_{p95}$ trends present a sparse and weak signal in the study domain, except for the winter (Fig. 3f). The southern coast presents a significant increase in the $Hs_{p95}$ values, which are greater than 2 cm/yr in some locations during the winter- 180 time. When looking at the mean $Hs$ trend, it is possible to see a general increase in this wave parameter along the Brazilian continental shelf, covering the coastal and offshore regions. The magnitude of the mean $Hs$ increase is small (< 0,2 cm/year) but significant in the whole period (Fig. 3b). The mean $Hs$ increase in winter is relatively greater (between 0.4 and 0.8 cm/year; Fig. 3g). The differences between the mean and $Hs_{p95}$ trends signal are in agreement with the findings of Young and Ribal (2019), who showed that the $Hs$ distribution changes in the last years were skewed to the left with an increase of small waves 185 - which can change the mean without changing the extreme percentiles. The trends in the number of extreme wave events also present sparse behaviour, but with significant increases along the Brazilian coast (Fig. 3c,h). The event's increase occurs on

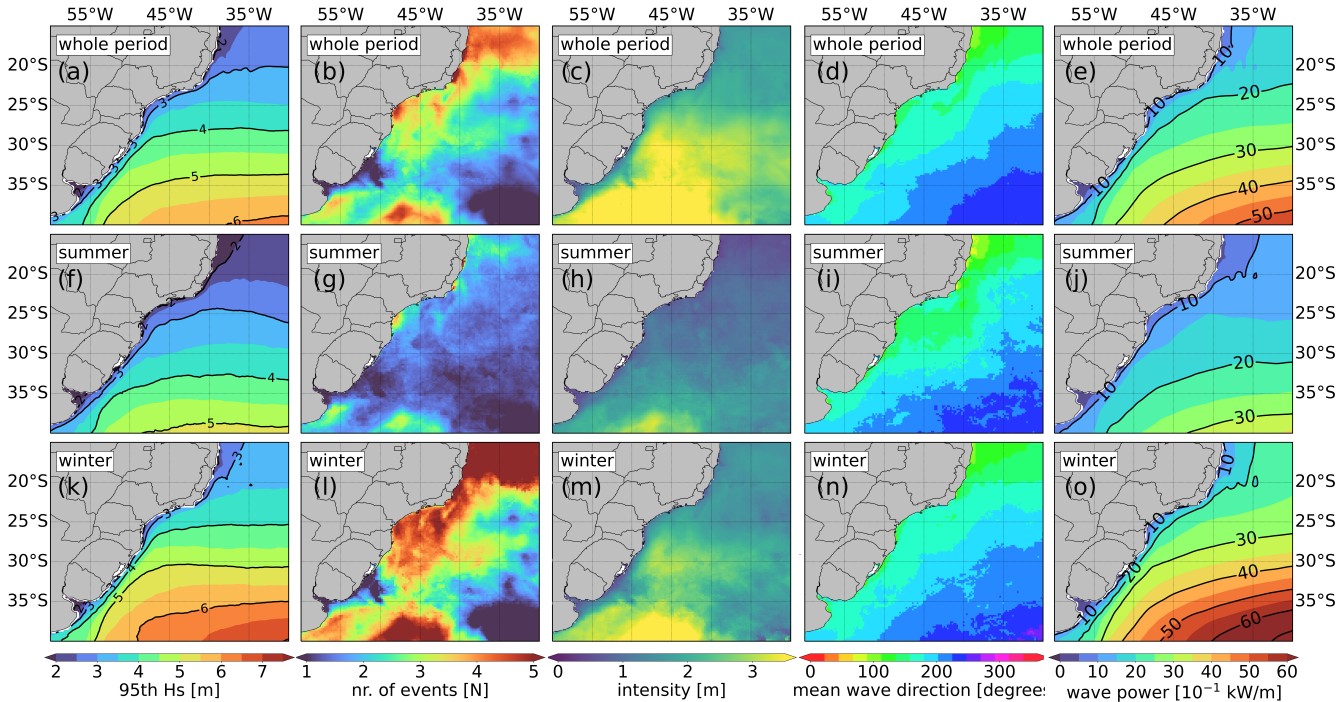

**Figure 2.** (a,f,k) Averaged monthly $95^{th}$-percentile $Hs_{peak}$ value [m], extreme wave event (b,g,l) number, (c,h,m) intensity ($Hs$ - $95^{th}$ $Hs_{peak}$) (m) (d,i,n) mean direction (degrees), and mean wave power [$10^{-1}$ $kW/m$] in the (a-e) whole period (1993-2021), (f-j) summer (DJF), and (k-o) winter (JJA) based on the CMEMS wave products (product ref. no. 1 and 2, Table 1)

most of the coast in the whole-period analysis, while it is confined to some portions of the southern and southeastern coast during the winter (Fig. 3h). It is important to note that the rise in the number of events does not follow the $Hs_{p95}$ trends pattern.

Figure 3 also shows the spatial trends of the mean distribution of the wave peak period and wave power during the events (Fig. 3d,e,i,j). There is a general increase in the peak period in the study domain, confined to the central portion of the coast in the winter (Fig. 3i). The increase in wave period and $Hs$ can lead to important changes in the wave power (Eq. 1). Note that for the wave power calculation, $Te$ was used and the trends presented in Fig. 3d,i are based on the $Tp$. However, considering a JONSWAP wave spectrum, $Te$ is directly related to $Tp$ ($Te = 0.9 \times Tp$; Guillou, 2020). The wave power presents a small but significant increase along the coast in the whole period and wintertime (Fig. 3e,j), reaching maximum values ($> 0,2$ kW/m/year) offshore southern Brazilian and Uruguayan coast (30 - 40°S) in the winter. Following the mean $Hs$ behaviour, the increase in the wave power is larger in the winter than in the whole period - as expected, once wave power is proportional to $Hs^2$ (Eq. 1). Other extreme event indicators, such as intensity, mean wave direction, and lifetime did not present a robust trend signal and, therefore, are not shown.

The extreme event analysis based on each grid point in a high-resolution hindcast provides a more detailed view of pattern changes along the coast. On the other hand, such an analysis can produce sparse results that may not be easily applied to

more practical and operational tasks. Therefore, the trends in some event parameters were analysed for each Brazilian Navy's monitoring and warning subareas (Fig. 4). For this analysis, we focused on the parameters with significant trends at least in one region and season, although both whole-period and winter trends are presented in Fig. 4 for consistency. Both C and D subareas present a significant increase in the number of events in the whole period. The trends of 0.2 and 0.28 events/year represent an increase of ~20% in the C and D subareas in 29 years (based on the increase of the annual mean of their series). Together with subarea B, these regions also present an increase in the mean wave power - despite no significant change in the peak period. In the winter, the A and C subareas present significant trends in the number of events per year, representing a 27.2% and 20% increase, respectively. C subarea presents a small but significant increase in peak period in the winter, as well as in the wave power. The wave power also increases in subarea D in the winter.

By the time series, it is possible to note a high interannual variability due to large-scale climate modes that affect the regional wave climate through storm track shifts (e.g., Sasaki et al., 2021). The SWSA is affected by many large-scale variability modes that interact, being widely studied in the atmosphere but not well understood in the wave fields (Godoi et al., 2020; Godoi and Júnior, 2020; Sasaki et al., 2021), which make it difficult to correlate climate indexes with wave parameters directly. However, even considering these variabilities, most parameters present a positive trend, although not always significant. As explained in Section 2.3, we consider the Mann-Kendall test to assess the significance of the trends. The sensitivity of the Mann-Kendall test may be related to the large variance of the time series, which directly influences the trend detected by this method (Wang et al., 2020).

## 3.3 Coastal risk analysis

The Charlies subarea (C) is one of the most affected locations, experiencing an increased number of extreme wave events, peak periods, and wave power in the last years. However, linking the changes in the regional wave climate with coastal hazards is not a straightforward task once the wave systems are modified by bathymetry, and their impact depends on the coastal morphodynamics. Table 2 presents the number of events recorded by the BDe-BS and the computed trends for each type of hazard. São Paulo coast was affected by 163 hazards between 1993 and 2021, of which 48% (78) were caused exclusively by storm waves and 30% (49) by the combination of waves and tides (either as a result of astronomical or meteorological tides). In the winter, 93% (51) of events are associated with waves, with combined events following a similar proportion (35%) of total events compared to the whole period (30%). Thus, the hazards forced exclusively by anomalous tides are rare in the winter, which may be related to the high wave events frequency in this season. The number of events on the coast increased both for the whole period and in winter (JJA). The increase in the number of coastal hazards was mainly led by wave events since events caused only by tidal influence did not present any significant trend. The results show an increase of 120% and 145% of total events and wave-forced events on the coast in 29 years, considering the whole-period mean. This high increase is in agreement with Souza et al. (2019), who found a pronounced increase in wave-forced hazards after the 2000s and 2010s decades [226% compared with the 1928-1999 period] when analysing a longer period of the same database (1928 - 2016).

Figure 5 shows the time series of the yearly events of coastal hazards from the BDe-BS against the spatial sum of the number of events in the C subarea obtained by the wave event analysis (described in Section 2.4). Due to the small number of coastal

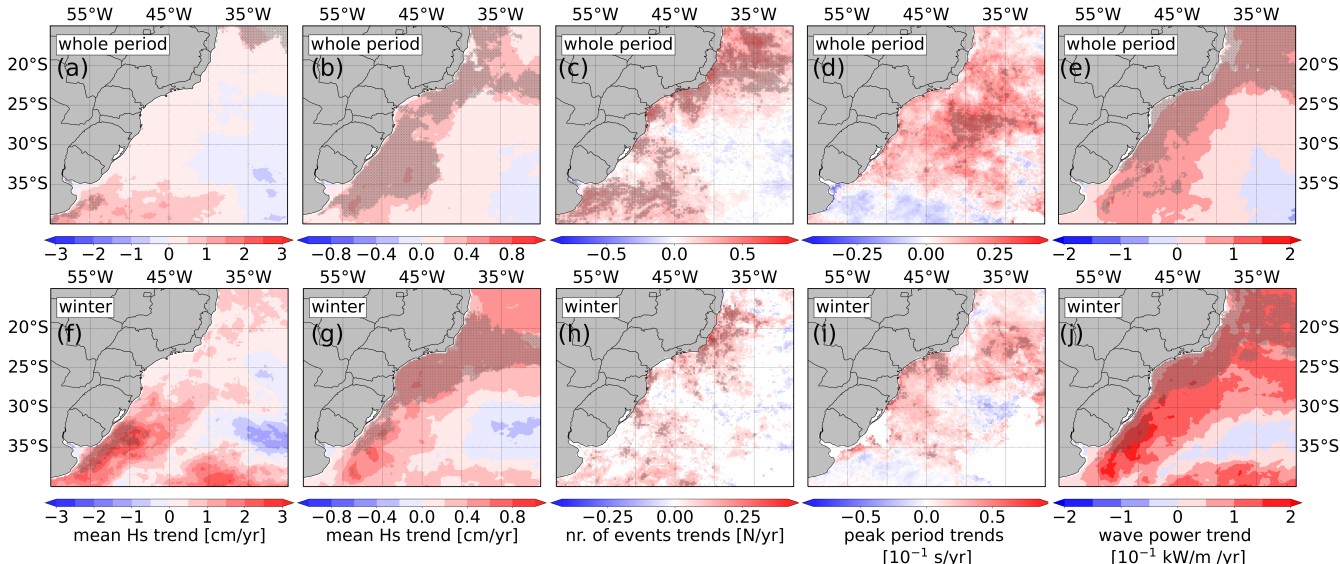

**Figure 3.** Trends in the (a,f) $95^{th}$-percentile $Hs_{peaks}$ [cm/year], (b,g) mean $Hs$ [cm/year], (c,h) number of extreme wave events [number/year], (d,i) peak period of the events [$10^{-1}$ s/year], and (e,j) mean wave power [$10^{-1}$ kW/m/year] in the (a-e) whole period and (f-j) winter (JJA) based on the CMEMS wave products (product ref. no. 1 and 2, Table 1). Grey crosses represent points where the trend is significant (Mann-Kendall test; p-value $\leq 0.05$).

hazards per year, we show the time series and trends for the whole period to obtain more robust statistics. Nevertheless, the winter trends presented the same signal and relation among the event types as the whole period trends (see Table 2). Note that the absolute numbers of events recorded on the coast and obtained in the C subarea are not comparable. This analysis aims to evaluate the similarity in the variability and trends of the time series. The number of events presented for the C subarea in Fig. 5 is the sum of events in all grid points within defined boundaries.

Most of the coastal hazards reported by the BDe-BS are associated with waves, combined or not with a tidal rise. This is revealed not only by the numbers in Table 2 but also in the time series (Fig. 5a). The trends of all events and events associated with waves are similar, especially considering the trend error. It is possible to compare the trends in the coastal hazards forced by waves (0.22 events/year, Table 2) with the trends of the number of extreme wave events in the C subarea (Fig. 5b; 0.20 events/year in Fig. 4a). However, considering the mean number of coastal events forced by waves (4.4 events/year), the
increase in the coast corresponds to 145% in 29 years.

Although there is no total agreement between the extreme events detected in the C subarea and hazards reported on the coast in the year-by-year analysis, the trend behaviour is similar. The number of combined events (wave + tides) and total events trend are superposed in Fig. 5b. These overlaid pictures show that differences in the wave-forced coastal hazards and extreme wave events in the C subarea may result from the influence of sea level elevation events forced by storms or astronomical tides.
For instance, 2002 and 2009 do not present a peak in the extreme wave events within the C subarea, but they were marked by a

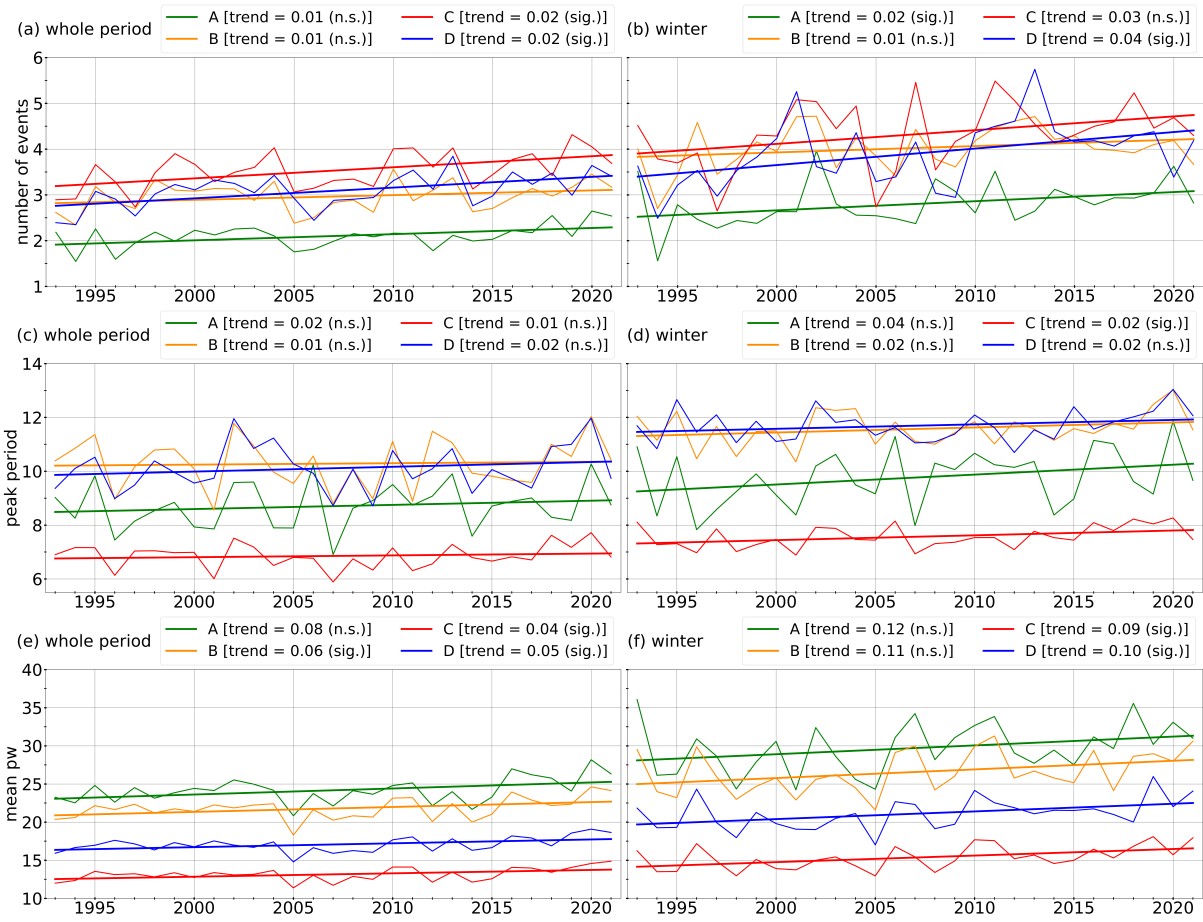

**Figure 4.** Annual time series and trends for the (a,b) number of extreme wave event [number/month] (c,d) mean peak period of the events [s] and (e,f) mean wave power [$10^{-1}$ kW/m] in the (a,c,e) whole period and (b,d,f) winter (JJA) computed for each Brazilian Navy's warning areas along the study domain's coast (Fig. 1) based on the CMEMS wave products (product ref. no. 1 and 2, Table 1). Trends units are in number/year, $10^{-1}$ s/year and $10^{-1}$ kW/m/year, respectively. N.S. and SIG. in the legend mean not significant and significant, respectively, according to the Mann-Kendall (p-value $\leq 0.05$).

high number of wave-forced coastal hazards related to a higher percentage of combined events. These wave events would not become a hazard if the local sea level elevation did not allow waves to reach further into the continental area. In addition, the disagreement between the coastal and offshore events time series can be addressed for bathymetry and morphology reasons.

Moreover, coastal hazards can also occur after a sequence of events that result in a more vulnerable coast due to the lack of recovery time (Souza et al., 2019). For instance, these additional elements to coastal erosion can explain the total increase of wave-forced events recorded on the coast in 29 years (145%) is much larger than the increase in the C subarea (~20%). In this case, the human use of modifying the shoreline may intensify the damage effects of the extreme wave events increasing (Muehe, 2018).

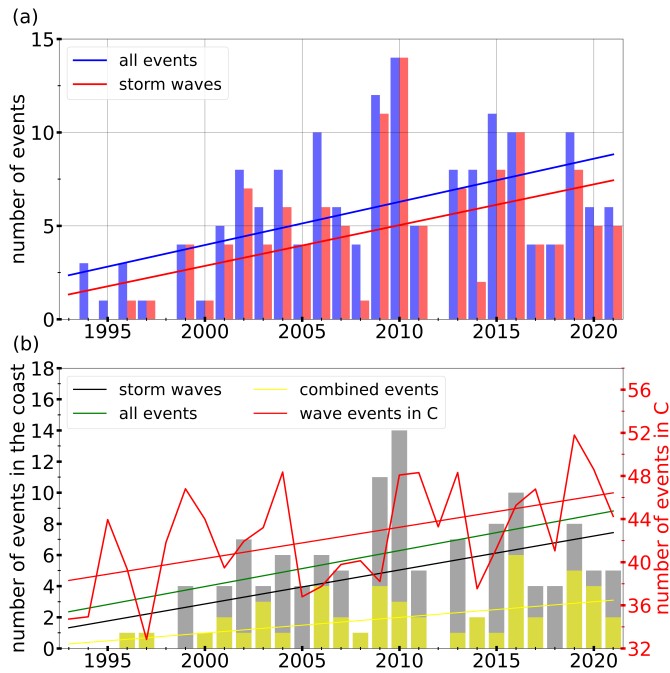

**Figure 5.** (a) Number of coastal hazards per year caused by any forcing (blue) and waves (red) based on BDe-BS (product ref. no. 3, Table 1); (b) Numbers and trends of coastal hazards per year associated with waves (grey bars and black line) and combined events (waves + tides; yellow bars and lines) based on BDe-BS and total number and trend of wave events in the subarea C (red lines) based on the CMEMS wave products (product ref. no. 1 and 2, Table 1). The trends presented in these panels are statistically significant (Mann-Kendall test; p-value $\leq$ 0.05), and the values are shown in Table 2.

## 4 Conclusion

The present work aimed to assess changes in the extreme wave climate in the SWSA, giving new insights into offshore coastal risk assessment and management in this domain. Understanding the extreme waves changes is crucial for supporting future projections, which are indispensable for the design and safety control of ship vessels, offshore and coastal structures and maintenance (e.g., oil/gas platforms, aquaculture, wind and wave farms), as well as coastal infrastructure (e.g., ports, roads, and touristic facilities) (e.g., Bitner-Gregersen et al., 2018; Vettor and Guedes Soares, 2020). Our findings showed important changes in the SWSA mainly associated with an increase in the mean $Hs$ values and wave period. These changes directly impact the offshore and coastal zone, increasing the wave power reaching the region and, consequently, aggravating the coastal hazards along the coast.

Even though extreme waves have a major role in coastal flooding and coastal erosion (e.g., Parise et al., 2009; Machado and Calliari, 2016), there still exists a lack of knowledge about how observed large to regional scale trends will affect the coast. Most of the Brazilian Navy's monitoring and warning subareas within MetArea V (WMO) require attention regarding wave climate changes. According to WAVERYS hindcast analysis, the number of extreme wave events (above the $95^{th}$-percentile

**Table 2.** The number of coastal hazards reported by the BDe-BS (product ref. no. 3, Table 1) in the whole period (1993 - 2021) and in the winter (JJA). The percentage calculation is based on the total number of events, which represents the sum of events forced only by waves, only by anomalous tides (either by astronomical or meteorological components) or by the combination of both ("waves + tides"). The last row highlights all events associated with waves, with or without tidal influence ("waves"). The trends unit is events per year/season, and bold values denote significance (Mann-Kendall test; p-value $\leq 0.05$).

| | whole period | | winter (JJA) | |
|---|---|---|---|---|
| | number of events | trend [number/year] | number of events | trend [number/season] |
| total | 163 (100%) | **0.23** | 55 (100%) | **0.14** |
| wave + tides | 49 (30%) | **0.10** | 19 (35%) | **0.06** |
| only waves | 78 (48%) | **0.12** | 32 (58%) | **0.06** |
| only tides | 36 (22%) | 0.01 | 4 (7%) | 0.01 |
| waves | 127 (78%) | **0.22** | 51 (93%) | **0.12** |

$Hs_{peaks}$) increased in the A, C, and D subareas and the mean wave power increased in the B, C, and D subareas. The trends vary depending on whether the whole period or only wintertime is considered. In this work, we analysed the winter (JJA) since it shows the most extreme wave patterns, but an extension of the analysis to other seasons is recommended for the future once the Autumn (MAM) and Spring (SON) weather patterns are also able to produce severe waves (e.g., Crespo et al., 2022). By our findings, we recommend special attention to C and D subareas once they present changes both in the number of events and wave power.

Regarding the coastal assessment, we found an increase in the number of coastal hazards in São Paulo State. According to our analysis, the increase in coastal hazards in this location is mainly associated with wave forcing and can be related to the increase in the number of extreme wave events in subarea C. Despite the well-known limitation in wave modelling, particularly to extreme waves (e.g., Campos et al., 2018), this finding gives evidence that the WAVERYS hindcast may be useful to assess, not only extreme wave climate in the study domain (as shown by Crespo et al., 2022) but also the events reaching the coast in a long to mid-term perspective. However, more care is needed for interannual and interseasonal analyses that require year-by-year assessment, especially because a coastal hazard depends not only on the waves. Sea level rise, in both climatic and synoptic scales, and astronomical tides play a large role, potentially turning moderate waves into damaged ones once a high sea level allows waves to propagate and break further in the continent. Souza et al. (2019) highlighted that the most severe coastal hazards reported in the region do not present the highest values of $Hs$ or sea level elevation but a combination of factors. Many other elements, such as coastal vulnerability, precipitation, morphology, and coastline orientation, affect the establishment of a coastal hazard (e.g., Muehe, 2018; Souza et al., 2019), mainly when the hazard is defined by its impact on the coast and not by some pure meteorological and/or oceanographic parameter.

Therefore, a complete assessment of coastal impacts needs more specific analysis considering local information and data, which is impracticable in this work, considering the study domain size. However, the trends derived herein are a valuable factor in identifying areas potentially vulnerable to climate change hazards and are also useful for engineers and stakeholders working towards the sustainable development of maritime activities. These changes may require adaptation measures, such as enhancing coastal protection (e.g., building dikes and harbours' protection measures). The findings reported in this work may also support the designing of new projects and future assessments that will allow the advance of the association of the large-scale wave climate with coastal impacts.

**Data and Code Availability**

The data products used in this article, as well as their names and documentation, are summarised in Table 1. The wave products are available through Copernicus Marine Service (https://marine.copernicus.eu/). The Baixada Santista Coastal Hazards database (BDe-BS) is available under request by email to Dr. Celia R. G. Souza (celia@sp.gov.br). Codes are available under request by email to the corresponding author.

*Author contributions.* CBG: Conceptualization, Formal analysis, Methodology, Visualization, Writing — original draft. JS: Conceptualization, Methodology, Writing — review & editing, Supervision. CRGS and PLS: Methodology, Writing — review. RC and PLSD: Writing — editing, Supervision.

*Competing interests.* The authors declare that they have no conflict of interest.

*Acknowledgements.* The authors would like to acknowledge Marcel Ricker for his support with the wave event method implementation. The study was supported by the European Green Deal project "Large scale RESToration of COASTal ecosystems through rivers to sea connectivity" (REST-COAST) (grant no. 101037097). We gratefully acknowledge the project DOORS (grant no. 101000518) and DAM Mission project CostalFuture. C.B.G. is funded by the Helmholtz European Partnership 'Research Capacity Building for healthy, productive and resilient Seas' (SEA-ReCap). This study also used data and resources from the projects: "Resposta Morfodinâmica de Praias do Sudeste Brasileiro aos Efeitos da Elevação do Nível do Mar e Eventos Meteorológico-Oceanográficos Extremos até 2100" (CAPES, proc. no. 88887.139056/2017-00), "Sistema de Aviso de Ressacas e Inundações Costeiras para o Litoral de São Paulo, com foco em Impactos das Mudanças Climáticas" (São Paulo Research Foundation (FAPESP), grant #2018/14601-0), and "Extreme wind and wave modelling and statistics in the Atlantic Ocean" (FAPESP, grants #2018/08057-5 and #2020/01416-0).

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
