# Peer review of "Recent changes in extreme wave events in the Southwestern South Atlantic"

_State of the Planet, 2022_

## Referee Comment (RC1)

**Review of "Recent changes in extreme wave events in the Southwestern South Atlantic"**

This study investigates the wave mean and extreme climate in the South West South Atlantic ocean (SWSA) and the potential associated coastal hazards from 1993 to 2021, focusing on a portion of the Brazilian coast. To do so, the author first used the outputs of regional wave datasets from the CMEMS and a historical coastal hazards dataset from the The Baixada Santista Coastal Hazards database (BDe-BS). The regional wave climate is investigated using several parameters (Significant wave height $H_s$, wave energy period Te, wave power, number of extreme events and intensity) for both mean regime and extreme wave climate (Hs $95^{th}$ percentile). The seasonality is also investigated with a particular attention on the winter period during which most intense wave events occur. Despite the limited period of study (29 years), significant and valuable results on trends are found for both the wave climate and coastal hazards, especially for the number of extreme events.

Overall, I found the general idea and workflow coherent, and the use of 2 different dataset very valuable. This study provides valuable results promoting the need of further research for coastal disaster prevention. Though, before publication, a significant number of clarifications on the methods is needed, extra efforts to link the wave climate to costal hazards is expected, along with an exhaustive review of the whole manuscript concerning Figures (both format and captions) and general writing.

For the following reasons, I reckon that the manuscript should be considered for publication after undergoing major revisions.

**General comments:**

- Although not a native speaker, I suggest checking for the whole manuscript for syntax and English writing. Also, many times there is a lack of connection and logic between statements (that alone are generally very true) within the same sentence. Though the article is fully understandable, the high number of technical corrections needed results in a major revision. Related comments are enumerated below in the technical and small corrections part (corrections carefully noted only until line 160 of the article).

- In general, there is lack of clarity on the CMEMS data used and on the methods used to compute for instance Hs $95^{th}$ percentiles (from hourly Hs time series or independent local events?), yearly time series of spatial average of specific areas (results showed in Figures 3 and 4), or trends from Sen's slope and parametric linear regression... Also, the comparison of the number of extreme events obtained from CMEMS database and BDe-BS (Figure 4c) is not very coherent, another analysis could be done, as further proposed. Related comments are chronologically presented in the Specific comments section. In the case all clarifications solve my doubts or misunderstandings, this would be a minor comment, otherwise a major revision is expected.

**Specific comments:**

1. Line 55-71, 2.1 Datasets: What is the point using these 2 CMEMS dataset? Is it because WAVERYS misses the year 2021? You state, "so a more consistent analysis can be achieved despite using different sources", and further compute statistical indicators (such as the 95$^{th}$ percentiles) with no explanation of the actual data used (Figure 1, caption: "based on the CMEMS hindcasts", what data are you plotting?). Furthermore, combining data from a 1) global reanalysis and a 2) global analysis and forecast from the same service (CMEMS) is a bit redundant.

2. It is not clear if you are using Hs independent events or hourly Hs time series when you compute $H_s$ 95$^{th}$ percentiles and the number of extreme events above the 95$^{th}$ percentiles. Line 75: "… the Hs peaks …", it could be easier to define a $Hs^{peaks}$ to further in the article refer to Hs independent events in an efficient way (e.g., line 77, "Hs distribution" is confusing), because percentiles computed from $H_s$ time series or $H_s$ peaks can give very different results… Another way would be to specify that further in the article, $H_s$ 95$^{th}$ values will refer to percentiles computed from $H_s$ independent events, while mean $H_s$ corresponds to average computed from hourly $H_s$ time series.

3. Lines 76-80: This part is not very clear, maybe you could give more details of the method used to eventually obtain the $H_s$ 95$^{th}$ values used in Figures 1,2 and 3. From what I understood, you first compute the monthly $H_s^{peaks}$ 95$^{th}$ percentile at every grid point of the study domain. From these values, the average seasonal and annual percentiles are computed (here, you could precise how you defined your seasons, SON, DJF…). Finally, at every grid point, your time series (1993-2021, 29 values) are the yearly percentiles computed independently of the seasonal variability, and yearly percentiles for each season.

4. Lines 83-89: I understand the method using Sen's slope to find trends and if they are statistically significant with a Mann-Kendall test. You have a trend +- error, and that is significant or not. Then, it is not very clear why and what are doing by computing a parametric linear regression analysis. First, as you mention lines 197-198, it assumes a normal distribution, which is not true for extreme wave event distribution. Then, you are for instance applying a bootstrap method (n = 1000) to a set of 29 yearly values of Hs 95$^{th}$ percentiles. How do you compute de confidence intervals (CIs)? Figure 3 shows constants errors but Cis not constant, with a narrowing at mid-term of the whole period.

5. Line 97: Here, you explain how the duration of each event was computed, following *Weisse and Günther (2007)*. Line 48, in the introduction, you also mention the duration, but you never further investigate this characteristic!

6. Lines 98-100: You just defined the duration of each event, and the intensity associated to this event, that is the difference between the SET value and the maximum value during this event. Following *Weisse and Günther (2007)*, you should clarify that you will then use the mean intensity, i.e., the average of the intensity of all individual events (above SET) within a year, both independently of seasonal variability and for each season.

7. Line 182-185: "The extreme event … warning subareas (Fig. 3)." I assume that you want to point out the benefits and limitation of such approach to perform coastal hazards assessment, to introduce the complementary focus on specific subareas (A, B, C, D) where you have a "terrain" database from the BDe-BS. Instead of "Due to that", I think it should be "Therefore". Moreover, it is not because of reanalyses derived-sparse results that you analyse the trends in A, B, C, D (previous results do not show higher statistical significance there), but because you have data there!! Finally, for each area, I assume that you work with spatially averaged trends (you should also explain it along with the associated spatially averaged errors), but what do you mean by "most relevant trends"? You compute spatial trends for each area and parameter and then find if they are statistically significant, but ALL trends are analysed. It is therefore quite hard to understand the reasoning here and I recommend modifying these lines to be clearer.

8. Lines 213-231: The trends calculated (from the reanalyses and the BDe-BS) are indeed similar in the subareas C. But as you say, yearly number of events from both databases often differ with a clearly higher number of wave event for the reanalyses-derived data than coastal hazards from the BDe-BS. This result raises the same concern than the comment 35 ("because percentiles computed from $H_s$ time series or $H_s$ peaks can give very different results"). Let's assume that for the number of extreme wave events over the Hs $95^{th}$ percentile, you used the distribution independent events (2 days' time-window), $H_s^{peaks}$. Therefore, in theory, the maximum number of independent events within one year is roughly 183. This is the distribution of wave independents events. However, you are working with EXTREMES wave events corresponding to events with Hs values higher than the SET value (Defined by the $95^{th}$ percentile). So, from 183 events, 5% are above de $95^{th}$ percentile, ≈ 9 extreme event per year at most, when Figure 4c shows that the number of extreme wave events ranges from 23 to 36. So, it means that, though you used the independent events distribution ($H_s^{peaks}$) for the $95^{th}$ percentile Hs values, the number of extreme events was computed from the $95^{th}$ percentile of hourly Hs time series.
So, to conclude: 1) I think that it is confusing and, in agreement with comment 35, you should clarify whether Hs hourly time series are used or $H_s^{peaks}$. 2) except for trends, the number of extreme events obtained from the 2 the reanalyses and BDe-Bs are not comparable with this method. For instance, if you would chose the $99^{th}$ percentile instead of the $95^{th}$, the number of extreme wave events detected would be lower and more like the number of extreme events given by the BDe-BS (you actually say that you could use any percentile between 90 and 99, Lines 94-95). In my opinion, I would only compare trends with this method. However, it could be interesting to assess the capacity of the reanalyses to reproduce extreme wave events that are documented by the BDe-Bs, giving an idea of the typical wave forcing leading to coastal hazards in the subarea C. I am aware of the extra work that it represents, but it is an opportunity to make the link between "the regional wave climate and with coastal hazards" (Line 201).

9. Lines 223-231: On the assumptions of differences observed between reanalyses-derived extreme wave events and coastal hazards from the BDe-BS. "Maybe these

wave events would not become a hazard if the local sea level rises did not allow waves to reach further into the continental area": this is indeed a potential consequence of sea level rise. However, the hypothesis of sea level rise explaining the difference observed only from the absence of peaks in 2002 and 2009 in the reanalyses-derived number of extreme wave events is off topic. It is a long and large-scale effect that has a low impact one year to another. Also, it occurs over the whole period, not only during two isolated years, and the provided data here to make such assumption is simply not enough. Finally, the number of extreme events is not necessarily related to the intensity of these events. A year with a low number of extreme wave events but powerful ones can lead to a higher number observed extreme event at coast than a year with a high number of relatively weaker extreme wave events.

10. The results show significant and high trends. Line 220: for the subarea C, maybe you could take advantage of giving an example of the consequence over 29 years, in order to give more weigh to your findings. For instance, 0.20 event/year increase looks small, but over 29 years it represents 5-6 more events per year in the subarea C that shows between 24-34 events per yaer (~20% increase !!).

**Figures comments:**

Overall, all figures need to be modified. First, I suggest removing Figure 4.a, and adding as a first Figure a plot that describes the study area. For instance, you could provide a Figure showing the whole area where you are using CMEMS data, and within this plot add a zoom on the A, B, C, D zones. It is a bit disturbing to provide a figure of the area of study at the end of the paper. Further comments on figures:

1. Figures 1,2,3: Overall, you have a high number of subplots. To gain space, when the x,y label and ticks are common to several subplots, only show it on the edges, same for the colorbars. This way you can narrow down the space between subplots. You may then increase the fontsize of labels and ticks (with same size for all, e.g., coordinates have not the same size). Also, putting directly on the figure "whole period, summer, winter" make it easier. For instance, you could have something like the following figure:

[Figure]

2. The colorbar of Mean wave direction (Figure 1) should be the circular (e.g. "hsv")
3. I suggest to modify the legend or Figure 1 " Mean of annual (a,e,i) 95th-percentile Hs (m) values (a,e,i) , extreme wave event (b,f,j) number, (c,g,k) mean intensity (Hs - 95th Hs) (m) (d,h,l) mean direction (degrees)  for the (a-d) whole period of study (1993-2021), … "
4. Line 185: 1) Figure 3 misses 2 subplots (95th percentile). 2) For the whole paragraph describing results from Figure 3, I suggest adding "(see Fig Xx)" for every observation, e.g., Line 187 "In the winter, the A and C subareas presented significant trends in the number of events per year (see Fig. 3b)." 3) Hs 95[th] trends are not described, probably because the corresponding figures are missing.

**Technical and small corrections:**

This is not an exhaustive list of needed corrections. Because of not being a native speaker, some comments/corrections might not be adequate, but still the manuscript should be carefully reviewed.

1. Line 4-5: "... the occurrence of  extreme wave events but also other extreme wave indicators that may impact the  offshore and coastal areas." Extreme wave indicators don't "impact" offshore and coastal areas, extreme waves do. Rephrase with something like "… but also extreme wave indicators that characterise the potential wave impact/hazards on offshore and coastal areas."
2. Line 5-8: "For a more direct application … impact the coastal zone". More direct than what? I suggest moving this sentence after the part describing the CMEMS data you

are using, since you first perform a regional assessment of the wave climate, and then link your findings with coastal hazards referenced by the BDe-BS.

3. Line 5-6: need to rephrase "For a more direct … a more focused …" because it is repetitive.

4. Line 9: it would be nicer to write $H_s$ instead of Hs for the whole paper, as well as $95^{th}$ instead of 95th.

5. Line 10: "… statistics are  performed to analyse  wave mean and extreme climate patterns …". I would rather write "mean and extreme wave climate", "mean wave climate", "extreme wave climate" here and then.

6. Line 13: if $H_s$ values, then wave period values, wave power values … For instance: "… in mean values of Hs and wave period, and consequently wave power.

7. Line 24: I suggest removing "thus reflecting directly on hazards along the coast" because it is a bit a standard phrase that is not needed.

8. Line 25: extreme wave  and storm surges occurrence?

9. Line 26-27: Rephrase "the SWSA … exploration fields". "Which is" refers to the SWSA, as if the ocean is responsible for the transportation … For instance, rephrase: "… where 755 million tons of goods were transported…"

10. Line 30: order chronologically the references (ICMBio, 2018; Pereira-Filho et al., 2021); Check the whole manuscript (e.g., Line 243).

11. Lines 33-35: "The difficulties remain mostly in the still-limited knowledge and understanding of the local physical processes and climate variabilities …" What unknown local physical processes can prevent from assessing wave trends? What climate variabilities? Following "for several reasons.", the reader expects a clear explanation or enumeration, but the reasons given are very vague.

12. Line 36: remove "In general" or "most"

13. Line 37: "… this  tendency"?

14. Line 39: "… present  larger uncertaintyies."

15. Line 40: In addition to understanding the significant wave height $(H_s)$ trends

16. Line 42: "… at both  regional …"

17. Line 43-44: Specify the period you are mentioning. Do you mean the changes between the present climate conditions and future scenarios? Or it is the changes that already occurred under the present climate (1993-2021), and other changes in the future (e.g., towards the end of the century)?

18. Line 45: "… coastal risk assessments …". The connection between the two parts of the sentence is not very clear to me and vague. What do you mean by linking the regional to local wave extremes? Regional scale would be the SWSA and local scale an area like you define later (A to D)? or beach scale? "… future scenarios … demand special efforts …" Why, knowing the future wave climate, would it be harder to link the regional and local wave extremes than for the present climate?

19. Line 47: "the recent wave climate", specify the exact period of study

20. Line 50-51: same comment than for 3.

21. Line 55: "global  reanalysis ", you have the wave reanalysis WAVERYS and a global analysis and forecast product (GLO-NRT)

22. Line 55-57: For the reader, it is a good idea to sum up the products used in a table. Though, too little information is given. I suggest including the main characteristics of the CMEMS product, such as temporal and spatial resolution, temporal coverage, wind forcing, and available data for the BDe-BS (the table indicates the period  1928-2021)

23. Line 57: write 1/5º instead of 0.20º to compare with the following 1/12º resolution.
24. Line :60 "Both products are produced  using the Météo France Wave Model"
25. Line 61: "WAVERYS is forced by hourly? surface winds and sea ice fraction fields …"
26. Line 65: "… western South Atlantic  wave climate…" as said in Crespo etal. (2022).
27. Line 65: "The authors compared  Hs …"
28. Line 67: "… against wave buoy measurements  at three locations…"
29. Line 68-69: "The ERA5 performance in representing the winds is also relevant once this field dominates the wave generation process", I don't get the point of this sentence, ocean surface wave generation is always initially driven by winds. Do you mean that the ERA5 performance is relevant for young-sea state while not crucial for swells that travelled away from their generation area?
30. Line 70: I suggest removing "even", because it suggests that it would be okay if it did not reproduce storm variability, or that is something not expected.
31. Line 78: "… next section …", aren't you referring to section 2.4 instead of the next section (2.3)?
32. Line 103, suggestion: change the section title to "Extreme wave event analysis"
33. Lines 106: Wave power: you are using the mean wave period $T_e$ directly available from CMEMS; In Figure 2, peak period ($T_p$)trends are showed; You should use the same. If you are using the $T_p$, the wave energy flux P (equation 1) is a bit different (see *Guillou, 2019, https://doi.org/10.1016/j.renene.2020.03.124*). For instance, using $T_p$, you should replace in equation (1) $T_e$ by $\alpha T_p$, where $\alpha = 0.9$ for a standard JONSWAP wave spectrum.
34. Line 106: Same comment as for 5., Hs,Te --> $H_s$, $T_e$.
35. Line 114: Specify that MetArea V corresponds to Atlantic waters west of 20ºW from 35º50'S to 7ºN.
36. Line 117: the results found here to improve future monitoring and warning system development .
37. Line 120: I suggest changing "the hemerographic method" for "a hemerographic literature review".
38. Line 122: Remove "Thus …", what is said in the previous phrase does not imply the definition of a coastal hazard.
39. Line 123: , the latter.
40. Line 137: "… controlled by the storm track due to the strongest winds associated with the cyclones", I suggest rephrasing "… controlled by cyclones storm tracks and associated strong winds"
41. Line 138: "The main storm track position, between … characterises", I suggest rephrasing "The storm tracks that are mainly confined between … characterise"
42. "… wave height distribution  with a Hs gradient towards the south …"
43. Line 142: "storm track**s**".
44. Lines 140-144: I think you should first present the results and then provide an explanation about the differences observed, otherwise it is a bit confusing. 1) what differences are observed between winter and summer (direction, 95th Hs, nr of events and intensity) and 2) why? (Southward shift of storm tracks).
45. Line 145: correction needed "… while spring (SON) remains the winter pattern"
46. Line 146: need to rephrase "For these reasons we are going to focus on the analysis of the winter since it heads the top list of extreme wave events in the study domains"

47. Line 153: Fig 1 b,j
48. Line 160: I suggest removing "Starting from the basic extreme statistics …"
49. Line 160: "present", use the same tense for the whole paper. You could sometimes use "show" for instance instead of always using "present"
50. Line 168: same as line 160.
51. Line 175: As you did earlier for the Hs trends, and since you observe a small but significant wave power increase, you could give the magnitude of a significant wave power increase.
52. Lines 180-181: aren't statistics more robust when the trend is significant? Whether for the Hs 95$^{th}$ or the number of events plots, the trend is overall significant at a higher of location for the whole period than for the winter season.
53. Line 190 "… high interannual variability"
54. Line 197-198: "… a normal distribution required by the parametric test." Then again, what is the point using parametric linear regression analysis?
55. Line 203-204: the percentage values are not coherent "… 48% (78) … 30%(49) …"
56. Line 205: What do you mean by "… the same proportion (35%) of total events", same than for the whole period?
57. Line 206: "High  wave events frequency"
58. Line 207: "Table 1 presents …" move this sentence before starting to mention data from the Table 1 (before Line 203)
59. Line 219-220: " …" remove this because you directly compare both trends! Also, since you have the 0.20 positive trend from Fig. 3a results, you can indicate the trend value for coastal hazards data.
60. Lines 243-244: First and second parts of the sentence are understandable alone, but lacks connection, I suggest rephrasing.
61. Line 248: You did not elect the winter season as representative of more extreme wave climate, you focused on winter because it shows the most extreme wave patterns!

---

## Author Response (AR1)

**Reply to Reviewer #1**

Dear reviewer,

Thank you for your comments and suggestions and for the time dedicated to this detailed revision. We improved the manuscript according to them, and we accepted the suggestions to improve the figures. Below are the responses to the major comments in red. At the end of this document, you will find the reply for the technical corrections in blue.

**Review of "Recent changes in extreme wave events in the Southwestern South Atlantic"**

This study investigates the wave mean and extreme climate in the South West South Atlantic ocean (SWSA) and the potential associated coastal hazards from 1993 to 2021, focusing on a portion of the Brazilian coast. To do so, the author first used the outputs of regional wave datasets from the CMEMS and a historical coastal hazards dataset from the Baixada Santista Coastal Hazards database (BDe-BS). The regional wave climate is investigated using several parameters (Significant wave height Hs, wave energy period Te, wave power, number of extreme events and intensity) for both mean regime and extreme wave climate (Hs 95th percentile). The seasonality is also investigated with a particular attention on the winter period during which most intense wave events occur.
Despite the limited period of study (29 years), significant and valuable results on trends are found for both the wave climate and coastal hazards, especially for the number of extreme events.
R. This manuscript is submitted for this special issue of the State of The Planet as part of the Copernicus Marine Service (CMEMS) Ocean Science Report 7. As a request, we need to use CMEMS products, including 2021. Therefore, the study is restricted to 29 years. This is better explained in the specific comments below.

Overall, I found the general idea and workflow coherent, and the use of 2 different dataset very valuable. This study provides valuable results promoting the need of further research for coastal disaster prevention. Though, before publication, a significant number of clarifications on the methods is needed, extra efforts to link the wave climate to costal hazards is expected, along with an exhaustive review of the whole manuscript concerning Figures (both format and captions) and general writing.
R. Thank you for your comment. The method was clarified in the specific comments below. We also improved the discussion about the links between the wave climate and coastal hazards.

For the following reasons, I reckon that the manuscript should be considered for publication after undergoing major revisions.

**General comments:**

- Although not a native speaker, I suggest checking for the whole manuscript for syntax and English writing. Also, many times there is a lack of connection and logic between statements (that alone are generally very true) within the same sentence. Though the article is fully understandable, the high number of technical corrections needed results in a major revision. Related comments are enumerated below in the technical and small corrections part (corrections carefully noted only until line 160 of the article).

R. Many thanks for your efforts in correcting these technical and small mistakes. We implemented your suggestion and checked the writing and editing to ensure a better manuscript. We accepted most of your suggestions pointed out as small corrections, and there were made directly in the reviewed manuscript.

- In general, there is lack of clarity on the CMEMS data used and on the methods used to compute for instance Hs 95th percentiles (from hourly Hs time series or independent local events?), yearly time series of spatial average of specific areas (results showed in Figures 3 and 4), or trends from Sen's slope and parametric linear regression... Also, the comparison of the number of extreme events obtained from CMEMS database and BDe-BS (Figure 4c) is not very coherent, another analysis could be done, as further proposed. Related comments are chronologically presented in the Specific comments section. In the case all clarifications solve my doubts or misunderstandings, this would be a minor comment, otherwise a major revision is expected.

R. We improved the explanation about the methods, focusing on the questions raised by you. Further explanations are in the specific comments section below.

**Specific comments:**

1. Line 55-71, 2.1 Datasets: What is the point using these 2 CMEMS dataset? Is it because WAVERYS misses the year 2021? You state, "so a more consistent analysis can be achieved despite using different sources", and further compute statistical indicators (such as the 95th percentiles) with no explanation of the actual data used (Figure 1, caption: "based on the CMEMS hindcasts", what data are you plotting?). Furthermore, combining data from a 1) global reanalysis and a 2) global analysis and forecast from the same service (CMEMS) is a bit redundant.

R. The use of CMEMS data, including 2021, is a requirement for the submission to the CMEMS Ocean State Report 7. In this way, we used the wave reanalysis of CMEMS (WAVERYS; Law-Chune et al., 2021) from 1993 to 2020 and added 2021 from the CMEMS Global Ocean Waves Analysis Near Real-Time product. The analysis is made as they are a consistent dataset from 1993 to 2021. We clarified this aspect in the reviewed manuscript.

Lines 56 - 59: *"The main dataset used in this work was the Copernicus Marine Service (CMEMS) global hindcast, named WAVERYS (Table 2, Ref. No. 1; Law-Chune et al., 2021), available from 1993 to 2020. To include 2021 in the analysis, the WAVERYS was complemented with data from the CMEMS Global Ocean Waves Analysis Near Real-Time product (GLO-NRT; Table 2, Ref. No. 2). The combination (in time) of these two products is referred to hereafter as CMEMS wave products."*

2. It is not clear if you are using Hs independent events or hourly Hs time series when you compute Hs 95th percentiles and the number of extreme events above the 95th percentiles. Line 75: "... the Hs peaks ...", it could be easier to define a Hspeaks to further in the article refer to Hs independent events in an efficient way (e.g., line 77, "Hs distribution" is confusing), because percentiles computed from Hs time series or Hs peaks can give very different results... Another way would be to specify that further in the article, Hs 95th values will refer to percentiles computed from Hs independent events, while mean Hs corresponds to average computed from hourly Hs time series.

R. We computed 95th percentiles of significant wave height (Hs) from Hs-independent events using a time window of 48 h, i.e., a minimum distance of 16-time steps between the peaks. We adopted $Hs_{peaks}$

instead of simply Hs to clarify the method, as you suggested. You can find a better explanation in the reviewed manuscript.

Lines 76 – 81: *"In this work, the percentiles were computed using the empirical distribution of the Hs peaks ($Hs_{peaks}$) within a given period, thus allowing us to obtain a more detailed view of individual wave events' occurrence. The selected $Hs_{peaks}$ must be separated by a minimum of 48 hours to guarantee the independence of the peaks. This time window has been widely applied in past studies to ensure the collection of one peak per storm (e.g., Caires and Sterl, 2005; Meucci et al., 2020). Besides that, 48 hours is a suitable but not-so-restrictive time threshold for extreme wave analysis in the region, particularly considering the differences among the seasons."*

3. Lines 76-80: This part is not very clear, maybe you could give more details of the method used to eventually obtain the Hs 95th values used in Figures 1,2 and 3. From what I understood, you first compute the monthly Hspeaks 95th percentile at every grid point of the study domain. From these values, the average seasonal and annual percentiles are computed (here, you could precise how you defined your seasons, SON, DJF...). Finally, at every grid point, your time series (1993-2021, 29 values) are the yearly percentiles computed independently of the seasonal variability, and yearly percentiles for each season.

R. Thank you for your comment. It is correct what you have understood. In the revised version we provided more clear explanation of the method on percentiles, We used the averaged monthly 95th percentiles as the base for all percentiles fields presented in the manuscript. These monthly percentiles were used to build the annual (12 months) and seasonal (3 months) mean and trends. For the application of the Weisse and Günther (2007) method, we need a unique 95th percentile field to establish a SET value per grid point. Since all your trend analysis is based on the monthly 95th percentile, we used the average of these monthly values over the whole period (29x12 months) as SET.

Lines 81 - 86: *"The 95th percentile is computed based on the monthly $Hs_{peaks}$ distribution in each grid point. Using these monthly 95th percentiles, we calculate the annual, seasonal, and whole-period means used for the trend and extreme events analysis (section 2.4). The seasonal mean of the 95th monthly percentiles is computed for the summer and winter, using the average December-January-February and June-July-August, respectively, thus having one value per year. The annual percentiles are computed by the average of all monthly percentiles within the year. A whole-period average of the monthly 95th percentile is used as a reference for the wave event analysis (section 2.4)."*

Lines 98 - 99: *"Moreover, the use of averaged monthly percentile results in a smoothed field, especially due to the $Hs_{peaks}$ variability among the year. In this way, for some locations, the exceedance of events above SET is large than 5%".*

4. Lines 83-89: I understand the method using Sen's slope to find trends and if they are statistically significant with a Mann-Kendall test. You have a trend +- error, and that is significant or not. Then, it is not very clear why and what are doing by computing a parametric linear regression analysis. First, as you mention lines 197-198, it assumes a normal distribution, which is not true for extreme wave event distribution. Then, you are for instance applying a bootstrap method (n = 1000) to a set of 29 yearly values of Hs 95th percentiles. How do you compute de confidence intervals (CIs)? Figure 3 shows constants errors but Cis not constant, with a narrowing at mid-term of the whole period.

R. We agree with you. We removed the parametric linear regression in the revised version of the manuscript since it presumes a normal distribution, which does not fit our analysis.

5. Line 97: Here, you explain how the duration of each event was computed, following Weisse and Günther (2007). Line 48, in the introduction, you also mention the duration, but you never further investigate this characteristic!
R. Thank you for your comments. We removed this part from the methodology section since the results did not show this parameter.

6. Lines 98-100: You just defined the duration of each event, and the intensity associated to this event, that is the difference between the SET value and the maximum value during this event. Following Weisse and Günther (2007), you should clarify that you will then use the mean intensity, i.e., the average of the intensity of all individual events (above SET) within a year, both independently of seasonal variability and for each season.
R. The reviewed manuscript better describes of the parameters of the wave, such as intensity, mean wave direction, etc.

Lines 100 - 104: *"Following Weisse and Günther (2007), the intensity is equal to the difference between the maximum Hs of the event and the SET at that point. The wave event statistics, such as the number of events, intensity, mean wave direction, and peak period, are presented herein as annual and seasonal means to build the spatial distribution and trends and obtain the spatial-averaged time series. The intensity and wave parameters were calculated by averaging all individual events (above SET) within the year or season."*

7. Line 182-185: "The extreme event ... warning subareas (Fig. 3)." I assume that you want to point out the benefits and limitation of such approach to perform coastal hazards assessment, to introduce the complementary focus on specific subareas (A, B, C, D) where you have a "terrain" database from the BDe-BS. Instead of "Due to that", I think it should be "Therefore". Moreover, it is not because of reanalyses derived- sparse results that you analyse the trends in A, B, C, D (previous results do not show higher statistical significance there), but because you have data there!! Finally, for each area, I assume that you work with spatially averaged trends (you should also explain it along with the associated spatially averaged errors), but what do you mean by "most relevant trends"? You compute spatial trends for each area and parameter and then find if they are statistically significant, but ALL trends are analysed. It is therefore quite hard to understand the reasoning here and I recommend modifying these lines to be clearer.
R. Thank you for your comment and suggestion. By using "show most relevant trends", we meant to show the parameters that presented a significant trend in at least one region and season. We modified the term and improved the explanation in the reviewed manuscript.

Lines 201 - 203: *"Therefore, the trends in some event parameters were analysed for each Brazilian Navy's monitoring and warning subareas (Fig. 3). We focus this analysis on the parameters that had significant trends at least in one region and season, although both whole-period and winter are presented in Fig. 3 for consistency."*

8. Lines 213-231: The trends calculated (from the reanalyses and the BDe-BS) are indeed similar in the subareas C. But as you say, yearly number of events from both databases often differ with a clearly

higher number of wave event for the reanalyses-derived data than coastal hazards from the BDe-BS. This result raises the same concern than the comment 35 ("because percentiles computed from Hs time series or Hs peaks can give very different results"). Let's assume that for the number of extreme wave events over the Hs 95th percentile, you used the distribution independent events (2 days' time-window), Hspeaks. Therefore, in theory, the maximum number of independent events within one year is roughly 183. This is the distribution of wave independents events. However, you are working with EXTREMES wave events corresponding to events with Hs values higher than the SET value (Defined by the 95th percentile). So, from 183 events, 5% are above de 95th percentile, ≈ 9 extreme event per year at most, when Figure 4c shows that the number of extreme wave events ranges from 23 to 36. So, it means that, though you used the independent events distribution (Hspeaks) for the 95th percentile Hs values, the number of extreme events was computed from the 95th percentile of hourly Hs time series. So, to conclude: 1) I think that it is confusing and, in agreement with comment 35, you should clarify whether Hs hourly time series are used or Hspeaks. 2) except for trends, the number of extreme events obtained from the 2 the reanalyses and BDe-Bs are not comparable with this method. For instance, if you would chose the 99th percentile instead of the 95th, the number of extreme wave events detected would be lower and more like the number of extreme events given by the BDe-BS (you actually say that you could use any percentile between 90 and 99, Lines 94-95). In my opinion, I would only compare trends with this method. However, it could be interesting to assess the capacity of the reanalyses to reproduce extreme wave events that are documented by the BDe-Bs, giving an idea of the typical wave forcing leading to coastal hazards in the subarea C. I am aware of the extra work that it represents, but it is an opportunity to make the link between "the regional wave climate and with coastal hazards" (Line 201).

R. (1) The SET applied in this study is based on the average of the monthly 95th percentile for the whole period (29 years x 12 months). This threshold is fixed for evaluating the trends throughout the period (as also explained in comment 3). The averaging smoothed the percentile, particularly due to seasonal variability among the year. The result is the pattern shown in Fig. 1a, which presents lower values than the percentile computed using the Hs$_{peaks}$ in a 29-years time series. The exceedance of the average percentile is larger than 5% for some locations. The choice of using e of the average of the monthly percentiles is to ensure consistency in the analyses. Otherwise, we would have several period-based percentiles, and the evaluation of the results would be inconsistent. We clarified the use of the averaged monthly 95th percentile in lines 98 - 99, as presented in comment 3. (2) The number of events in subarea C is not comparable with the events in the coast due to several factors, such as wave direction, storm clustering, storm tide. Some waves above the percentile in subregion C do not reach the coast and/or do not have the right incidence direction to penetrate the shoreline with strength enough to be recorded as extreme coastal events. This matter is also complemented in comment 9. The absence of a straightforward agreement is also a valid result in our manuscript since we discuss these several factors, including the higher increase trend in the coastal events. We added this discussion to the reviewed manuscript.

Lines 255 – 258: *"For instance, these additional elements to coastal erosion can explain the total increase of wave-forced events recorded on the coast in 29 years (145%) is much larger than the increase in the C subarea (20%). In this case, the human use of modifying the shoreline may intensify the damage effects of the extreme wave events increasing (Muehe, 2018)."*

9. Lines 223-231: On the assumptions of differences observed between reanalyses-derived extreme wave events and coastal hazards from the BDe-BS. "Maybe these wave events would not become a

hazard if the local sea level rises did not allow waves to reach further into the continental area": this is indeed a potential consequence of sea level rise. However, the hypothesis of sea level rise explaining the difference observed only from the absence of peaks in 2002 and 2009 in the reanalyses-derived number of extreme wave events is off topic. It is a long and large-scale effect that has a low impact one year to another. Also, it occurs over the whole period, not only during two isolated years, and the provided data here to make such assumption is simply not enough. Finally, the number of extreme events is not necessarily related to the intensity of these events. A year with a low number of extreme wave events but powerful ones can lead to a higher number observed extreme event at coast than a year with a high number of relatively weaker extreme wave events.

R. By "sea level rise", we meant the sea level elevation due to the storm or astronomical tides. However, we understand that using this term may be confusing in this context and we changed it to sea level elevation. This is explained further in the revised manuscript to avoid misunderstandings (lines 247 and 250). In this way, we addressed the differences in the peaks and troughs of the oceanic and coastal series to the coupled effect of storm waves and storm tides. In some situations, energetic waves can arrive on the coast, but a low sea level does not allow the penetration of the wave further into the continent to cause enough damage or erosion, thus not being included in The Baixada Santista Coastal Hazards database (BDe-BS). In this case, the effect of sea level elevation can be caused both by astronomical and atmospheric tides since the database does not differentiate them (Linhares et al., 2021).

10. The results show significant and high trends. Line 220: for the subarea C, maybe you could take advantage of giving an example of the consequence over 29 years, in order to give more weight to your findings. For instance, 0.20 event/year increase looks small, but over 29 years it represents 5-6 more events per year in the subarea C that shows between 24-34 events per year (~ 20% increase !!).

R. Thank you for your suggestion. In the revised version, we added a better description of the results in Fig. 3, highlighting the consequences of these trends in 29 years:

Lines 205 - 208: *"The trend of 0.2 and 0.28 events/year represents an increase of ~20% in the C and D subareas in 29 years (based on the increase of the annual mean of their series). Together with subarea B, these regions also showed an increase in the mean power wave despite no significant change in the peak period. In winter, the A and D subareas demonstrate significant trends in the number of events per year, representing a 27.2% and 37% increase, respectively."*

Lines 229 - 230: *"The results show an increase of 120% and 145% of total events and wave-forced events on the coast in 29 years, considering the mean over the whole period."*

Lines 243 - 246: *"It is possible to compare the trends in the coastal hazards forced by waves (0.22 events/year, Table 1) with the trends of the number of extreme wave events in the C subarea (Fig. 4c); 0.20 events/year in Fig. 3a). However, considering the mean number of coastal events forced by waves (4.4 events/year) the increase in the coast corresponds to 145% in 29 years."*

Lines 255 – 258: *"For instance, these additional elements to coastal erosion can explain the total increase of wave-forced events recorded on the coast in 29 years (145%) is much larger than the increase in the C subarea (~20%). In this case, the human use of modifying the shoreline may intensify the damage effects of the extreme wave events increasing (Muehe, 2018)."*

*References:*

Caires, S., and A. Sterl. "100-Year Return Value Estimates for Ocean Wind Speed and Significant Wave Height from the ERA-40 Data", *Journal of Climate* 18: 1032-1048, https://doi.org/10.1175/JCLI-3312.1. 2005

Law-Chune, S., Aouf, L., Dalphinet, A., Levier, B., Drillet, Y., and Drevillon, M.: WAVERYS: a CMEMS global wave reanalysis during the altimetry period, Ocean Dynamics, 71, 357–378, https://doi.org/10.1007/s10236-020-01433-w, 2021.

Linhares, P. S., Fukai, D. T., and Souza, C. R. G.: Clima de ondas e maré em três eventos meteo-oceanográficos extremos ocorridos em São Paulo, em fevereiro e abril de 2020, In: X Congresso sobre Planejamento e Gestão das Zonas Costeiras nos Países de Expressão Portuguesa, APRH/ABRhidro, 06-10/12/2021 (on-line), [in Portuguese], 2021.

Meucci, A., Young, I. R., Hemer, M., Kirezci, E., and Ranasinghe, R.: Projected 21$^{st}$-century changes in extreme wind-wave events, Sci. Adv., 6, eaaz7295, https://doi.org/10.1126/sciadv.aaz7295, 2020

Weisse, R. and Günther, H.: Wave climate and long-term changes for the Southern North Sea obtained from a high-resolution hindcast 1958–2002, Ocean Dyn., 57, 161–172, https://doi.org/10.1007/s10236-006-0094-x, 2007.
* * *
**Reply to supplementary comments**

**Figures comments:**

Thank you very much for your attention to the Figures. We adjusted them according to your suggestions. The actions are listed below.

- Figure 1 is now the study area.
- We removed common x,y labels and ticks to gain space.
- We increased and uniformed the fonsize.
- We included "whole period", "winter" and "summer" directly in the figures.
- The mean wave direction now has a circular colorbar.
- We adjusted the legends accordingly.

**Technical and small corrections:**

This is not an exhaustive list of needed corrections. Because of not being a native speaker, some comments/corrections might not be adequate, but still the manuscript should be carefully reviewed.
R. Thank you for the time you spent listing these corrections. We corrected and reviewed the manuscript and accepted most of the suggestions. The specific comments are replied below and marked in blue in the reviewed manuscript.

1. Line 4-5: OK.

2. Line 5-8: OK.

3. Line 5-6: OK.

4. Line 9: We changed the 95th to $95^{th}$. However, we rather use italic to refer to $Hs, Tp$, and $Te$ variables since some of them have an additional subscript, e.g., $Hs_{peaks}$.

5. Line 10: OK.

6. Line 13: OK.

7. Line 24: We would rather keep the phrase.

8. Line 25: OK.

9. Line 26-27: OK.

10. Line 30: Thank you. We checked the references.

11. Lines 33-35: OK.

12. Line 36: OK.

13. Line 37: OK.

14. Line 39: OK.

15. Line 40: OK.

16. Line 42: OK.

17. Line 43-44: OK.

18. Line 45: OK.

19. Line 47: OK.

20. Line 50-51: OK.

21. Line 55: OK.

22. Line 55-57: Thank you for the suggestion, but the Table 2 format follows the requirement of the special issue to the CMEMS Ocean Science Report 7.

23. Line 57: OK.

24. Line:60 OK.

25. Line 61: OK.

26. Line 65: OK.

27. Line 65: OK.

28. Line 67: OK.

29. Line 68-69: We rewrote to be more clear.

30. Line 70: OK.

31. Line 78: OK.

32. Line 103: Thank you, we accepted your suggestion.

33. Lines 106: We used the Te to calculate the wave energy, as presented in the methods. However, we rather to show Tp in the trend maps since it is a more well-known variable. The discussion relating Tp trends with the increasing wave power is still valid since Tp is a fraction of Te, as we explained in your comment. We commented on this relation in the manuscript (lines 192-194).

34. Line 106: We rather to use the mathematical form (in italics) to refer to these variables since some of them have an additional subscript, e.g., $Hs_{peaks}$.

35. Line 114: OK.

36. Line 117: OK.

37. Line 120: We would rather keep the term "hemerographic method" since it is the term used in the original database reference (Souza et al., 2019).

38. Line 122: OK.

39. Line 123: OK.

40. Line 137: The term storm track (singular), as used in the manuscript, refers to the region where the storms are more likely to occur (e.g., see Hoskins and Hodges, 2005 and AMS glossary, https://glossary.ametsoc.org/wiki/Storm_track). In the manuscript, we want to clarify that we are referring to the mean trajectory and not to individual storm tracks.

41. Line 138: Same as for 40.

42. OK

43. Line 142: Same as for 40.

44. Lines 140-144: We rewrote to be more clear.

45. Line 145: OK.

46. Line 146: OK.

47. Line 153: OK.

48. Line 160: OK.

49. Line 160: OK.

50. Line 168: OK.

51. Line 175: OK.

52. Lines 180-181: We removed this phrase.

53. Line 190: OK.

54. Line 197-198: As explained in the major comments, we removed the parametric test.

55. Line 203-204: OK.

56. Line 205: We rewrote to be more clear.

57. Line 206: OK.

58. Line 207: OK.

59. Line 219-220: OK.

60. Lines 243-244: We rewrote to be more clear.

61. Line 248: OK.
* * *
**Reply to Reviewer #2**

Dear Dr. Alejandro Orfila,

Thank you very much for your comments and suggestions. We improved the manuscript according to them, and we specifically addressed the raised points below in red.

Review Ms sp-2022-7  ''Recent changes in extreme wave events in the Southwestern South Atlantic'' by Gramcianinov et al.

The Ms. studies the spatial and temporal variability in the extreme wave events in the Southwest South Atlantic using CMEMS global wave reanalisis and near real time products for the period 1993-2021. Authors analyze the annual and seasonal Hs, extreme wave events defined as the 95th percentile of Hs, peak period, intensity,  wave direction and wave power providing insights on how trends would impact the coastal zone using a coastal hazards database. The paper is well written, the methods sound,  and the research provides a good overview of the long term evolution of extreme wave events in the area of the SWSA.  Prior publication however some minor points should be addressed.

1. Why is the duration for the extreme events set as 48h? Usually a 72h period is considered for an independent analysis of storms.

R.   In our method to compute the percentile, the minimum storm duration is 48 h, but it can persist for longer. The Hs peaks selection is made in two steps: 1) all peaks in the time series are selected by local maxima by comparing neighbouring values; 2) smaller peaks are removed until the minimum distance condition is fulfilled. We choose 48 h based on references (e.g., Caires and Sterl, 2005; Meucci et al., 2020), as it is a not-so-restrictive time threshold, especially to mid-latitudes, considering seasonal analysis. Pick a large time threshold would result in higher percentiles that may hinder the extreme event analysis in some locations and seasons. This is explained better in the revised manuscript.

Lines 77 – 81: *"The selected $Hs_{peaks}$ must be separated by a minimum of 48 hours to guarantee the independence of the peaks. This time window has been widely applied in past studies to ensure the collection of one peak per storm (e.g., Caires and Sterl, 2005; Meucci et al., 2020). Besides that, 48 hours is a suitable but not-so-restrictive time threshold for extreme wave analysis in the region, particularly considering the differences among the seasons."*

2.   Regarding #1. Is the intensity and number of events robust in front of the time window?.
R. To respond to this comment, we present two points:
(a) The number of events is indeed high if we consider the 5% most intense events using a time window of 48 h. The SET applied in this study is based on the average of the monthly 95th percentile for the whole period (29 years x12 months) since we need a fixed threshold for the period to evaluate the trends. The averaging smoothed the percentile, mainly due to seasonal variability throughout the year. The result is presented in Fig. 2a, which shows lower values than the percentile computed if we used the $Hs_{peaks}$ in a 29-years time series. The exceedance of the average percentile is larger than 5% for some locations. The choice of using e of the average of the monthly percentiles is to ensure consistency in the analyses. Otherwise, we would have several period-based percentiles, and the evaluation of the results would be inconsistent. We clarified the use of the averaged monthly 95th percentile in the reviewed manuscript.

Lines 98 – 99*: "Moreover, the use of averaged monthly percentile results in a smoothed field, especially due to the Hspeaks variability among the year. In this way, for some locations, the exceedance of events above SET is large than 5%."*

(b) The number of events and intensities are comparable to previous studies, even though method differences exist, thus making a straightforward comparison difficult. For instance, Gramcianinov et al. (2021) used the 90th percentile and a vary time window computed according to the autocorrelation function in each grid point. They found that the mean of 1.3 and 5.5 extratropical cyclones per year promoted extreme waves event in the region in the summer and winter, respectively. These values are coherent with those presented in the maps of Fig. 1f,j. Regarding the intensity, the same authors found the mean Hs of 6.5 m associated with the cyclones' events, which is also comparable to the intensity values (above the percentile) in some locations of the study domain (Fig. 1g, k). Moreover, Machado et al. (2010) accessed extreme wave events in the coastal region between 30ºS and 32ºS and found a mean of 1.33 events per year above the 90th percentile (1979-2008). We also reported this relatively small value at this exact location in Fig. 1b,f,j. We added some information about the robustness of our findings compared to previous studies in the reviewed manuscript.

*Lines 166 – 175: "The overall pattern and values presented in Fig. 2 agree with previous studies, even though method differences exist, thus making a straightforward comparison difficult. For instance, Gramcianinov et al. (2021), using the 90th percentile computed through a spatially-varying time window, found a mean of 1.3 and 5.5 extratropical cyclones per year associated with extreme waves event in the region in the summer and winter, respectively. These values are comparable with the number of events presented in the maps of Fig. 2b,g, l. Regarding the intensity, the same authors found the mean Hs of 6.5 m associated with these cyclones' events, which is also comparable to the intensity values (above the percentile) in some locations of the study domain (Fig. 2c,h,m). Moreover, Machado et al. (2010) accessed extreme wave events in the coastal region between 30ºS and 32ºS and found a mean of 1.33 events per year above the 90th percentile between 1979 and 2008. We also reported this relatively small value at this exact location in Fig. 2b,g,l. In this way, the method applied herein presents robust results according to what is reported in the region."*

3. Figure 2. If possible I suggest to include in this figure the wave power due to the additional interest in potential locations for power generation.
R. Thank you for your suggestion. In the revised version, we added the mean wave power maps in Figure 2 (e,j,o).

4. Figure 3. The caption does not correspond to the Figure.
R. We revised and adjusted the mistakes in the captions.

5. Figure 3 a,b, c and d. Besides the trend, can there be any relation in the wave climate and extreme events inferred from the large scale climatic modes of variability? (see for instance https://doi.org/10.1016/j.pocean.2021.102660).
R. Thank you for this comment. It is difficult, indeed, to directly relate the trend in the SWSA with climatic modes since there are many regional-to-large scale interactions affecting the region (e.g., ENSO, PSA, MJO). These modes interact with each other in different time scales resulting in different outcomes for the storm tracks and, consequently, for the waves. In this way, it is not trivial to correlate wave variability and climatic index or separate their effect over the region. A fully dedicated study needs to be conducted to address this. We added a discussion about this problem in the revised version of the manuscript.

*Lines 210 – 213: "By the time series, it is possible to note a high interannual variability due to large-scale climate modes that affect the regional wave climate through storm track shifts (e.g., Ramos et al., 2021; Sasaki et al., 2021). The SWSA is affected by many large-scale variability modes that interact, being widely studied in the atmosphere but not well understood in the wave fields (Godoi et al., 2020; Godoi and Torres Júnior, 2020; Sasaki et al., 2021), which make it difficult to correlate climate indexes with Hs parameters directly."*

The paper is a very nice contribution.
Thank you.

**References:**

Caires, S., and A. Sterl. "100-Year Return Value Estimates for Ocean Wind Speed and Significant Wave Height from the ERA-40 Data", *Journal of Climate* 18: 1032-1048, https://doi.org/10.1175/JCLI-3312.1, 2005

Godoi, V. A. and Júnior, A. R. T.: A global analysis of austral summer ocean wave variability during SAM–ENSO phase combinations, Climate Dynamics, 54, 3991–4004, https://doi.org/10.1007/s00382-020-05217-2, 2020.

Godoi, V.A., de Andrade, F.M., Durrant, T.H. *et al.* What happens to the ocean surface gravity waves when ENSO and MJO phases combine during the extended boreal winter?. *Clim Dyn* **54**, 1407–1424 (2020). https://doi.org/10.1007/s00382-019-05065-9

Gramcianinov, C. B., Campos, R. M., de Camargo, R., and Guedes Soares, C. Relation Between Cyclone Evolution and Fetch Associated With Extreme Wave Events in the South Atlantic Ocean. ASME. *J. Offshore Mech. Arct. Eng.*; 143(6): 061202. https://doi.org/10.1115/1.4051038, 2021.

Machado, A. A., Calliari, L. J., Melo, E., and Klein, A. H.: Historical assessment of extreme coastal sea state conditions in southern Brazil and their relation to erosion episodes, Panam. J. Aquat. Sci., 5, 105–114, 2010.

Marília S. Ramos, M. S., Farina, L., Sérgio Henrique Faria, S. H., Li, C. Relationships between large-scale climate modes and the South Atlantic Ocean wave climate, *Progress in Oceanography*, 197, https://doi.org/10.1016/j.pocean.2021.102660, 2021.

Meucci, A., Young, I. R., Hemer, M., Kirezci, E., and Ranasinghe, R.: Projected 21[st]-century changes in extreme wind-wave events, Sci. Adv., 6, eaaz7295, https://doi.org/10.1126/sciadv.aaz7295, 2020

Sasaki, D. K., Gramcianinov, C. B., Castro, B., and Dottori, M.: Intraseasonal variability of ocean surface wind waves in the western South Atlantic: the role of cyclones and the Pacific South American pattern, Weather Clim. Dyn., 2, 1149–1166, https://doi.org/10.5194/wcd-2-1149-2021, 2021.